# Acute Responses to Low and High Intensity Exercise in Type 1 Diabetic Adolescents in Relation to Their Level of Serum 25(OH)D

**DOI:** 10.3390/nu12020454

**Published:** 2020-02-11

**Authors:** Artur Myśliwiec, Maria Skalska, Beat Knechtle, Pantelis T. Nikolaidis, Thomas Rosemann, Małgorzata Szmigiero-Kawko, Agnieszka Lejk, Joanna Jastrzębska, Łukasz Radzimiński, Dorota Wakuluk, Karolina Czapiewska, Guillermo F. López-Sánchez, Zbigniew Jastrzębski

**Affiliations:** 1Department of Physiology, Gdansk University of Physical Education and Sport, 80-336 Gdansk, Poland; admysliwiec@wp.pl (A.M.); lukaszradziminski@wp.pl (Ł.R.); wkrecik@gmail.com (D.W.); czapiewska.karolina@wp.pl (K.C.); zb.jastrzebski@op.pl (Z.J.); 2Department of Pediatrics, Diabetology and Endocrinology, Gdansk Medical University, 80-210 Gdansk, Poland; mariajastrzebska@gumed.edu.pl (M.S.); mkawko@gumed.edu.pl (M.S.-K.); agnieszka.lejk@op.pl (A.L.); joanna.jastrzebska@hotmail.com (J.J.); 3Institute of Primary Care, University of Zurich, 8091 Zurich, Switzerland; thomas.rosemann@usz.ch; 4Medbase St. Gallen Am Vadianplatz, 9001 St. Gallen, Switzerland; 5Exercise Physiology Laboratory, 18450 Nikaia, Greece; pademil@hotmail.com; 6Faculty of Sport Sciences, University of Murcia, 30003 Murcia, Spain; gfls@um.es

**Keywords:** type 1 diabetes, serum 25(OH)D, oxygen consumption, blood glucose, exercise intensity

## Abstract

The main purpose of this study was to investigate the differences in glycaemic reaction in response to various physical activities in 20 young boys (14.4 ± 1.6 years) with type 1 diabetes mellitus (T1DM) and with either vitamin D deficiency or with suboptimal levels of vitamin D. Participants were divided into two groups (deficiency group—DG, *n* = 10; suboptimal group—SG, *n* = 10) according to their vitamin D levels. All patients performed aerobic and mixed (aerobic-anaerobic) physical efforts. During the exercise, the respiratory responses and glucose levels were monitored. Biochemical blood analyses were performed before each physical effort. The oxygen consumption was not significantly lower in SG during both aerobic and mixed effort (4.0% and 5.6%, respectively). The glycated haemoglobin (HbA1c) level was higher by 6.1% and the total daily dose of insulin (DDI) was higher by 18.4% in the DG. The differences were not statistically significant. Patients with lower vitamin D levels demonstrated an insignificantly higher glycaemic variability during days with both aerobic and mixed exercises. An appropriate vitamin D concentration in T1DM patients’ blood may constitute a prophylactic factor for hyperglycaemia during anaerobic training and hypoglycaemia during aerobic training.

## 1. Introduction

Type 1 diabetes mellitus (T1DM) is an autoimmune disease in which the insulin-producing beta cells of the pancreas are destroyed by effector lymphocytes sensitized to pancreatic antigens [1]. An increasing incidence of T1DM, observed in Poland during the last 20 years, has led to the fact that T1DM is the most common non-infectious chronic disease in the population under the age of 18 years in Europe. The highest incidence rate is reported in developing countries of Middle-East Europe, including Poland [2]. The data originating from different regional Polish registers, collected within the framework of the EURODIAB project over the last two decades, have demonstrated the dramatic increase of T1DM rate by 300%. In the light of epidemiological data from years 1989–2004, it is presumed that the incidence of T1DM will increase fourfold until 2025, particularly in the group of children at the age 5–9 [1].

On the basis of current scientific knowledge and numerous publications, it is confirmed that vitamin D plays a key role in T1DM pathogenesis. It has been proven that vitamin D deficiency (<75 mmol/L) constitutes a risk factor for diabetes and there is evidence suggesting that vitamin D deficiency may play a role in the development of pancreatic beta cells autoimmunity [3]. In addition, Yuan et al. [4] have proved that higher vitamin D concentrations are related with a lower risk of type 2 diabetes development, suggesting that vitamin D plays an important role in increase of insulin sensitivity in patients with carbohydrate metabolism disorder.

The aforementioned relation has been demonstrated by Zipitis et al. [5] in the EURODIAB study, who have emphasized the role of vitamin D supplementation in children in reducing the risk of T1DM development later in life. Given the currently available, however still scarce, studies it is highlighted that the development of T1DM may be correlated with the presence of VDR (Vitamin D receptors) in pancreatic beta cells. According to the current clinical and pre-clinical evidence, vitamin D may have a role in T1D prevention mainly through its immunomodulatory and anti-inflammatory effects rather than its effects on pancreatic beta-cell secretory capacity.

The vitamin D activity affects the expression of the VDR, and consequently, regulates the secretion and the activity of insulin [6]. Furthermore, it has been revealed that calcitriol is capable to transform proinsulin in to its active form-insulin [7]. Wimalawans et al. [8] have underlined the association between normal range vitamin D and calcium concentrations and the prevention of insulin resistance and T1DM. The emerging evidence shows that vitamin D, due to its role in immune system regulation (T regulatory cells), protects pancreatic beta cells, which are still not affected by the autoimmune process [9] therefore, it may prolong the remission period of T1DM [10]. What is more, Gabbay et al. [11], Aliabri et al. [12] and Panjiyar et al. [10] have revealed the favorable effect of optimal vitamin D level in patients with T1DM on metabolic control of the disease evaluated by glycated hemoglobin level (HbA1c), C-peptide concentration, insulin sensitivity and reduction of total daily dose of insulin.

The study of Al Zubeidi et al. [13], carried out in 2016, confirmed this assumption. These researchers showed a correlation between low serum vitamin D level and a higher incidence rate of diabetic ketoacidosis in patients with T1DM. On the basis of their studies, Hafez et al. [14] and Panjiyar et al. [10] recommend maintaining an optimal vitamin D level to mitigate the risk of hypoglycaemic events in this group of patients. The goals of T1DM therapy consist not only in achieving normal glycaemic levels and minimizing the risk of acute (i.e., hypoglycemia, diabetic ketoacidosis) and chronic complications (i.e., nephropathy, retinopathy, neuropathy and cardiovascular disorders), but also in preserving a normal functioning of patients in society, enabling them to lead the lifestyle approximated to that of their peers and to realize all of the current activities, including sport practice [13].

To date, no studies exist regarding the relationship between vitamin D level in diabetic patients and the tolerance of physical effort. Even though beneficial effects of physical effort are well known in the healthy population, recommendations regarding physical activity in patients with T1DM are essentially based on the results of few studies carried out in diabetic adults, frequently affected with type 2 diabetes [15,16,17].

The aim of this research was to reveal the differences in glycaemic reaction in response to aerobic and aerobic-anaerobic physical effort in young patients with T1DM diagnosed with vitamin D deficiency or having a suboptimal vitamin D level. Before proceeding to the study, we assumed that patients with higher vitamin D concentrations would demonstrate a better response to physical effort, measured by the amount of absorbed oxygen. Moreover, it was hypothesised that the period of hyperglycaemia and hypoglycaemia in this group would last shorter during the physical activity and after its completion.

## 2. Materials and Methods

### 2.1. Characteristics of the Study Group

The study group consisted of 20 boys with T1DM diagnosed according to the criteria of ISPAD guidelines [18], who remain under the care of the Clinic of Pediatrics, Diabetology and Endocrinology at the University Clinical Center in Gdańsk, Poland, a city located at latitude 54°22′ north. The mean age of patients was 14.4 ± 1.6 years and the mean duration of the disease was 6.7 ± 4.1 years. The body weight in the whole study group was 59.5 ± 12.8 kg and the mean BMI was 20.2 ± 2.6 kg/m^2^.

The protocol of the study was explained to every participant before the enrolment in the experiment. Patients and their parents signed the written informed consent form. Researchers obtained the approval of the Bioethical Commission of the Medical University of Gdansk (NKBBN/397/2018). Inclusion criteria of the research were: at least one year of T1DM duration. Patients using insulin pump therapy integrated with continuous glucose monitoring (CGM). Medtronic sensor and insulin pumps, Paradigm Veo and MiniMed 640G were used in the study. CGM measured real-time glycaemia and was not blinded, therefore the researchers and the patients had continuous access to glucose concentrations. In addition, inclusion criteria of the research were: assessment of the puberty at the stage II, IV or V in the Tanner scale, not practicing high performance sports, lack of diabetic ketoacidosis or severe hypoglycemia incidents during last five years and signed written consent form. The mean duration of continuous subcutaneous insulin therapy was 6.0 ± 3.9 years (*n* = 20). The mean HbA1c was 7.3 ± 0.8% (*n* = 20). Exclusion criteria were obesity, concomitant chronic diseases (e.g., hypothyroidism, liver, renal disorders and coeliac disease) that may have an impact on the occurrence of hypoglycemia, and the lack of written informed consent.

Patients did not practice any high-performance sports and their physical activity consisted of physical education classes at school and physical recreation, approximately 2–3 times a week. The group of 20 participants enrolled in the study was divided into two subgroups characterized by members with vitamin D deficiency (deficiency group <50 nmol/L, *n* = 10,) and patients with suboptimal vitamin D concentration (suboptimal group 50–75 nmol/L, *n* = 10) [19]. The participants did not receive any vitamin D supplementation throughout the year before the study and during the study period. The general characteristics of the study groups are presented in Table 1.

Basic physical examination was performed. All children underwent physical examination, and height and weight were taken in a standard way using Harpenden stadiometer and digital scale (Seca, Hamburg, Germany). BMI-z score (body mass index) SDS (standard deviation score of BMI, according to the formula: SDS-BMI = (BMI current − BMI 50 centile) / 0.5 (BMI 50 centile − BMI 3rdcentile)) was calculated with standard formulas, using the results of the OLAF study of Polish children [20].

### 2.2. Experimental Run

The experimental run consisted of three exercise tolerance tests. The objective of the first test was to determine the physical efficiency of every participant. The second test was based on 45 min of physical exercises, which intensity corresponded to aerobic metabolism and the third test at the intensity of aerobic–anaerobic metabolism (combined).

All patients ate the same meal (natural yoghurt, oat flakes, banana, walnuts) two hours before every of three exercise tolerance tests. It was composed of 60% of carbohydrates, 15% of proteins and 25% of fat. 40 g of natural yoghurt, 15 g of oat flakes, 40 g of unripe banana and 4 g of walnuts were included in two carbohydrates units (CU). The amount of CU depended on the total daily caloric requirement of the patient and participants received 0.7 to 0.8 units of insulin for every CU and every fat/protein unit (FPU).

Each participant received a bolus of rapid-acting insulin analogue before the meal, which did not affect glycaemia during the physical activity regarding the time of action of the drug. Blood glucose concentration did not exceed 8.32 mmol/L before enrolment to the aerobic physical exercise and 10 mmol/L before the commencement of the aerobic-anaerobic exercise. Glucose concentrations were controlled with the use of CGM, under medical supervision during every physical tolerance effort and one hour after its completion, and onwards up to 24 h after the training.

During the first experimental runs, all of the participants accomplished the progressive load physical tolerance test after preliminary medical procedures. On the basis of obtained results, researchers calculated the individual value of maximum oxygen consumption (VO_2_max) and anaerobic threshold (AT) rate expressed in power units on a bicycle ergometer. These ratios were essential to determine an individual load as a relative value of AT in subsequent physical tests. Two weeks after, every patient completed the first experimental 45 min aerobic physical activity after preliminary medical procedures. The load was expressed in power units (Watts) and the value was determined as 40% less than AT (Watts). After the next two weeks, patients were subjected to 30 min aerobic-anaerobic (combined) physical activity test, which consisted of two min of AT (Watts) minus 40% and 4 min of AT (Watts) plus 10% (Watts) alternately (5 repetitions altogether). Cardiopulmonary indices were continuously registered during both physical tolerance tests with the use of an expiratory gas analyzer.

### 2.3. Physiological Analyses

VO_2_max was measured with the use of expiratory gas analyser Oxycon Pro (Erich JAEGER GmbH, Hoechberg, Germany, 2012). Experimental runs were performed in physical effort laboratory in standard conditions (temperature 21 °C; atmospheric pressure 1010 hPa; air humidity 55%). Physical tolerance test was preceded by 5 min warm-up in the form of an ergometric work (Eos Sprint, Jeager, Hoechberg, Germany) with the load of 1 W/kg, at the rate of 60 rotations per minute. Subsequently, starting with the sixth minute of test, the load was increased every minute by 0.25 W/kg. The physical activity was interrupted when the rotation rate decreased by more than 10%, that is less than 54/min. The highest relative oxygen consumption, maintained for 15 s, at the end of the exercise was considered as VO_2_max. The anaerobic Threshold (AT) was determined by the analysis of expiratory gases exhaled during exercise tolerance test. AT was calculated as a quotient of carbon dioxide exhalation and oxygen consumption (respiratory exchange ratio—RER). When the calculated value was ≥1, it was assumed that the AT was reached. After experimental run patients rested sitting for 5 min and the expiratory gas analyzer was detached.

### 2.4. The Control of Blood Glucose Concentrations

Raw data from CGM were downloaded and trimmed to include only measurements from the exercise days. Only records with >70% of expected measurements (200 out of 288) were accepted for analysis. Gllycemic control parameters (daily mean glucose, standard deviation SD, coefficient of variance CV, times above (TAR), in (TIR) and below (TBR) 3.88–10 mmol/L range) were calculated with Glyculator 2.0. [21]. The insulin dose was assessed as daily insulin dose per kg of body weight (IU/kg).

### 2.5. Biochemical Analyses

Laboratory tests were performed in accredited Central Laboratory of University Clinical Centre in Gdańsk. The biological material for the study was venous blood collected from fasting patients just before the tests. HbA1c was determined by high-performance liquid chromatography (HPLC) using the Bio-Rad VARIANT™ HbA1c Program (Bio-Rad Laboratories, Inc., Hercules, CA, USA), with its values represented as percentages. Serum concentrations of 25(OH)D (reference range 50–75 nmol/L) were measured by chemiluminescence on Liason XL (DiaSorin, Stillwater, Oklahoma USA). The level of glycated haemoglobin (HbA1c, (<6.5%) was determined by high performance liquid chromatography (HPLC) using the Variantfrom BioRad. Total cholesterol (2.97–4.91 mmol/L), HDL cholesterol (>1.03 mmol/L) and triglycerides (<1.69 mmol/L) were measured on analyzers Alinity (Abbott, Wiesbaden, Germany) using the enzyme-linked immunoassay method, while LDL cholesterol (<115 mmol/L) was calculated using the Friedewald formula. The levels of thyrotropic hormone (TSH, 0.35–4.94 uU/mL) and free thyroxine (fT4, 9.01–19.05 pmol/L) were determined by a two-stage immunochemical method using microparticles and a chemiluminescent marker from Abbott, Germany. Alanine (<55 U/L), aspargine aminotransferase (5–34 U/L) and urine albumin (<20 mg/L) were determined by spectrophotometry using an Abbott Alinity analyser (Wiesbaden, Germany).

### 2.6. Statistical Analyses

The Shapiro–Wilk test was used to assess normal distribution of obtained data. In the case of normality, t-test for independent samples was applied for group comparison. The nonparametric U Mann–Whitney test was employed for variables with non-normal distribution. The statistical significance level was set at *p* < 0.05. Effect size (ES) for statistical differences was determined using Cohen’s *D* [22]. According to Hopkins et al. [23], the ES was classified as trivial (<0.2), small (>0.2–0.6), moderate (>0.6–1.2), large (>1.2–2.0) and very large (>2.0–4.0). All of the calculations were carried out using Statistica 13.0 software (Tulsa, OK, USA).

## 3. Results

Organism response to aerobic and aerobic-anaerobic (mixed) physical activity in the studied groups is presented in Figure 1. There were no statistically significant differences in VO_2_ between DG and SG groups during both exercise tolerance tests as well as within any phase of the tests. However, in the SG group, the VO_2_during the aerobic test was lower than in the DG group by ~4%. Although, the difference between the two groups was higher in case of mixed physical activity by ~5.6%. For every phase of the training, the magnitude effect was calculated in accordance with Hopkins et al. classification [23]. Its value was at the low level for every phase of aerobic physical activity, and in case of mixed physical activity, it was low and trivial. There were no differences in any biochemical blood parameter analysed in both groups of patients. HbA1c level was higher by 6.1% and a total daily dose of insulin (DDI) was higher by 18.4% in the DG group in comparison with the SG group. The differences were not statistically significant. In the DG group, TSH concentration was higher by 18.6% than in the SG group. TSH and FT_4_ levels, as well as albumin concentration in 24 h urine collection, were in the normal range and no statistically significant differences have been showed (Table 2).

During days with anaerobic training patients with 25(OH)D deficiency demonstrated insignificantly higher glycaemic variability measured with SD (*p* = 0.0603) and CV (*p* = 0.0746). This finding was reinforced by longer time spent in clinically significant hyperglycaemia >13.9 mmol/L (*p* = 0.1862), although these results did not reach statistical significance. Patients from DG group demonstrated insignificantly higher glycaemic variability measured with SD (*p* = 0.3218) and CV (*p* = 0.1761) during training with aerobic intensity. This finding was reinforced by longer time spent in clinically significant hyperglycaemia >13.9 mmol/L (*p* = 0.1864) as well hypoglycemia <3.88 mmol/L (*p* = 0.2414), although these results did not reach statistical significance (Table 3).

During experimental tolerance test with mixed intensity mean glucose concentrations in blood were higher in the DG group than in the SG group (7.9 ± 1.6 mmol/L versus 7.4 ± 1.5 mmol/L) (*p* = 0,5459). Glycaemic variability was also higher in the DG group, CV (31.3 ± 6.6% versus 26.1 ± 5.0%) (*p* = 0.074) and the period with persisting hyperglycaemia above 10.0 mmol/L (time above range—TAR), was longer (23.5 ± 17.3% versus 15.4 ± 21.8%) (*p* = 0.4055). In addition, the same group of patients was more prone to hypoglycemia during mixed training days (time below range < 3.88 mmol/L (TBR) 8.2 ± 7.6% versus 3.9 ± 5.5%) (*p* = 0.2414) in comparison with the days with the aerobic training.

## 4. Discussion

Twenty young boys affected with T1DM participated in the experiment. Due to different vitamin D levels, patients were divided into two subgroups, the DG group with vitamin D deficiency and the SG group with suboptimal vitamin D level. Both subgroups were homogeneous regarding biological development, biometrics and physical capacity (VO_2_max, VO_2_/AT) (Figure 1).

The experiment aimed to determine if vitamin D concentration has a significant impact on glycemic variability in relation to the type of training, with aerobic or anaerobic-aerobic intensity, undertaken by patients. Researchers decided to examine the issue, due to the lack of publications regarding the correlation between vitamin D level in children or adolescents with T1DM and effort tolerance.

In the view of results obtained in the experiment, vitamin D deficiency in patients affected with T1DM may have an unfavorable effect on organism response to the effort tolerance, in case of high as well as low-intensity physical activities. Furthermore, vitamin D deficiency may be one of the most significant factors responsible for glycaemic variability (CV%) towards both hyperglycaemia and hypoglycaemia. Additionally, it may prolong the time spent in hyperglycaemia (TAR) and hypoglycaemia (TBR) during aerobic and mixed training as well as 24 h after its completion.

The value of VO_2_max achieved in the study group does not seem to differ considerably from the group of healthy, not practicing individuals at the same age (Figure 1). The data regarding physical capacity in adolescents with T1DM are limited and the findings are equivocal. Cuenca-Garcia et al. [25] have shown the decreased oxygen capacity measured by VO_2_max in patients affected with T1DMin comparison to a healthy population. On the other hand, Adolfsson et al. [26] did not demonstrated substantial differences between such groups in the context of oxygen capacity in response to physical activity of different intensity. In the presented study researchers have not stated any relevant differences in VO_2_maxand VO_2_/AT between groups with lower and significantly higher vitamin D concentrations. According to the suggestions of Koundourakis et al. [27] and Jastrzębska et al. [28] vitamin D level may constitute one of the factors influencing substantially the oxygen capacity in young sports athletes. The volume of oxygen uptake in the DG and the SG group was presented in Figure 1. In spite of the fact that significant differences have not been found, patients with higher vitamin D levels consumed less oxygen during aerobic training by 4.0% and aerobic-anaerobic training by 5.6%. The fundamental goal of therapy in diabetes is the prevention of acute and chronic complications development. At present, theHbA1c level is used widely in order to evaluate this risk. In addition, an increasing number of data indicates that glycaemic variability is responsible for blood vessels damage in diabetic patients. It can be assessed thanks to better access to CGM. Lending support to this view is the fact that patients with well-controlled diabetes and low HbA1c levels nevertheless develop chronic and acute complications. Maiorino et al. [29] and Škrha et al. [30] have assumed that there are pathophysiological premises suggesting that glycaemic variability may cause damage to the wall of blood vessels, especially to the endothelium.

The excessive glucose concentration, that is accumulated abruptly in epithelial cells, cannot be metabolized by glycolysis within an adequately short period of time. It triggers the stimulation of additional metabolic pathways of glucose and increases the production of reactive oxygen species, which are toxic to the epithelium. If the energy influx continues to fluctuate rapidly and ensues the energy deficiency right after the energy excess (what happens in case of a sudden reduction in glycaemia), then intracellular metabolic mechanisms may dysregulate completely, leading as a consequence to early degeneration of the cell and its disintegration. Therefore, the factors affecting glucose concentration and glycaemic variability are the subject of ongoing studies.

In recent years, few studies focusing on the relationship of vitamin D level in T1DM children with the metabolic control of the disease, insulin sensitivity and development of acute vascular complications including hyperglycaemia and hypoglycaemia, have been published. It has been revealed that adequate vitamin D concentration has a beneficial effect on the reduction of the number of diabetic ketoacidosis [13,24] and hypoglycaemic [12,14] events requiring hospitalization. However, there are no studies regarding vitamin D level in paediatric population with diabetes, that take into account the physical activity and the type of training performed by patients.

The active lifestyle is important in the prevention and the treatment of many chronic diseases. The independent impact of physical activity on diabetes prevention is unknown, however, Petrie et al. have described its role as a key element of lifestyle modification [1]. Intensified and controlled physical activity in children with T1DM may improve glycaemic control, insulin sensitivity and prolong the period of remission in individuals with newly recognized T1DM. Moreover, it helps to prevent vascular disorders and ameliorates body composition, the quality of life and health throughout the whole life [2]. On the other hand, due to impairment of glycaemic control capacity, patients with T1DM are exposed to a higher risk of adverse effects like glycaemic variability towards hyper- and hypo-glycaemia, which can lead to acute, life-threatening complications during physical effort and chronic angiopathies in the future. Thus, in this group of patients, it is requisite to determine the risk factors of glycaemic variability during physical activity, considering the type of training as well.

The results of the present study showed that 20 T1DM patients had decreased vitamin D level (52.3 ± 24.4 nmol/L), which could have a negative impact on their metabolic control. After dividing the participants into two groups, the DG and the SG group, it was stated that the group of patients with vitamin D level <50 nmol/L was characterized by worse metabolic control in comparison with the group with suboptimal vitamin D concentration (HbA1c 7.6 ± 0.8% vs. 7.1 ± 0.8%) and required higher total daily dose of insulin, though the differences were statistically insignificant (Table 2).

Other researchers confirm this assumption and underline in their works the possibility of insulin resistance development in case of severe vitamin D deficiency [11,12,24]. Hitherto, only a sparse number of studies point out the higher risk of diabetic ketoacidosis [13] and the tendency to develop hypoglycaemia [14,24] in the group of T1DM patients with vitamin D deficiency. However, studies regarding the reaction of the organism to glucose concentrations and to glycaemic variability in this group of patients, which take into account the level of vitamin D and the type of performed training (high or low intensity), were not found in the available literature. In the presented study, the researchers have proven that T1DM patients with higher vitamin D levels (SG group) had lower glycemic variability (SD, CV, Table 3) during the mixed and aerobic training than the participants from the DG group.

Hence, the authors assume that the results presented in the experiment are innovative and have essential clinical and therapeutic implications. It is significant that the study was tailored and conducted in such a manner to exclude the other factors affecting glucose level during physical activity. The experiment was carried out during the same season, winter (in Northern Europe it is the time of the most severe vitamin D deficiency among its inhabitants), in the afternoon. The group consisted of participants of the same sex who performed the physical activity at the same level until the participation in the study. The tests were conducted after the consumption of the same meal, which constituted the first breakfast. The periods between the analogue of rapid-acting insulin administration, breakfast and the beginning of physical activity were equal. Glucose concentration before approaching the training was in the range of 8.32–10.0 mmol/L. In the study group, all of the other disorders that can affect glycaemia were excluded (Figure 1).

Preliminary results suggest that vitamin D deficiency in patients with T1DM who perform exercises at the intensity of aerobic energy transformations, cause high glycaemic variability towards both hyperglycaemia and life-threatening hypoglycaemia. It is noteworthy that in the group of patients who accomplished the training at the aerobic-anaerobic intensity, glucose concentrations were higher with the tendency towards hyperglycaemia. The authors of the study are aware of the fact that the group size was small. Although the results of the present study are interesting and essential in the light of clinical experience, they require the confirmation on larger group of patients with T1DM.

## 5. Conclusions

In conclusion, we assume that adequate vitamin D concentration in blood obtained by vitamin D supplementation in T1DM patients may constitute a preventive factor for hyperglycaemia during anaerobic training and hypoglycaemia during aerobic training. Moreover, it may avert the development of late-onset vascular complications by reduction of glycaemic variability in both types of physical effort.

## Figures and Tables

**Figure 1 nutrients-12-00454-f001:**
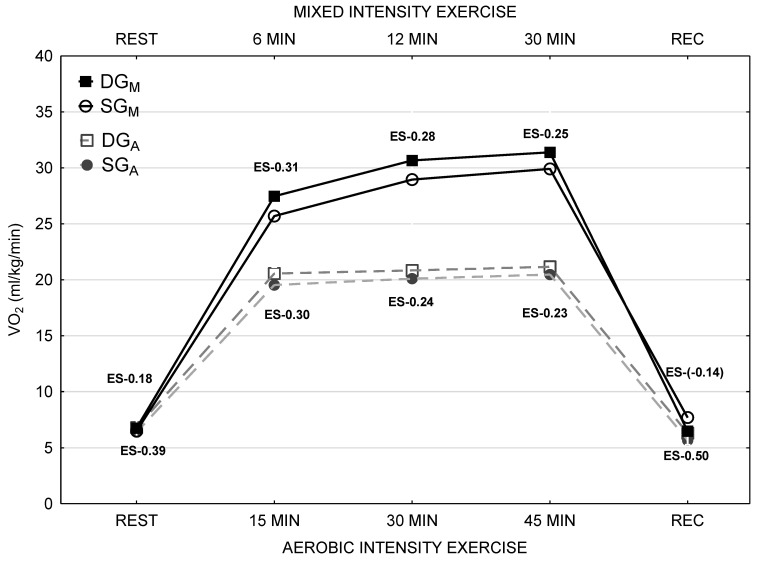
The volume of oxygen absorbed by patients from DG and SG groups during aerobic and mixed intensity exercise.

**Table 1 nutrients-12-00454-t001:** Characteristic of the participants.

Characteristics	Group DG	Group SG	*p*-Value
Age (years)	14.6 ± 1.6	14.6 ± 1.8	1.0001
Body height (cm)	170.5 ± 9.2	171.8 ± 13.3	0.8021
Body weight (kg)	59.4 ± 10.9	59.5 ± 15.0	0.9924
BMI (kg/m^2^)	20.4 ± 2.7	20.0 ± 2.5	0.7142
BMI percentile	51.5 ± 29.6	47.5 ± 23.6	0.7421
BMI z-score	0.20 ± 0.8	0.03 ± 0.7	0.6096
Disease duration (years)	7.4 ± 4.3	5.9 ± 4.0	0.4323
Insulin pump therapy duration (years)	6.8 ± 4.1	5.2 ± 3.7	0.3734
VO_2_max (ml/kg/min)	40.6 ± 5.1	39.9 ± 6.6	0.4532
VO_2_/AT (Watt)	27.3 ± 3.3	25.9 ± 4.2	0.7912
25(OH)D (nmol/L)	33.4 ± 10.3	64.8 ± 10.6	0.0002

**Table 2 nutrients-12-00454-t002:** Clinical and biochemical comparison of deficient (DG) and suboptimal (SG) 25(OH)D serum levels.

25(OH)D Status	DG (*n* = 10)	SG (*n* = 10)	*p*-Value
HbA1c (%)	7.6 ± 0.8	7.1 ± 0.8	0.2300
DDI (UI/kg)	1.0 ± 0.3	0.8 ± 0.3	0.1926
TSH (µU/mL)	1.6 ± 1.0	1.8 ± 0.5	0.7407
FT4 (pmol/L)	11.9 ± 1.2	11.9 ± 1.1	0.8200
ALAT (U/L)	21.3± 9.6	17.2 ± 4.4	0.3509
ASAT (U/L)	15.6± 4.0	14.3 ± 4.5	0.8003
Total cholesterol (mmol/L)	4.0 ± 0.7	4.2 ± 0.6	0.6903
HDL-cholesterol (mmol/L)	1.5 ± 0.4	1.6 ± 0.5	0.6685
LDL-cholesterol (mmol/L)	2.2 ± 0.5	2.2 ± 0.6	0.9494
Triglycerides (mmol/L)	0.8 ± 0.5	0.7 ± 0.3	0.7056
Urine albumin (mg/L)	6.3 ± 2.2	10.4 ± 4.8	0.1100

DDI—daily dose of insulin [24].

**Table 3 nutrients-12-00454-t003:** Comparison of glycaemic control parameters during days of aerobic and mixed exercise according to 25(OH)D deficiency.

Type of Exercise	Mixed			Aerobic		
25(OH)D Status	DG (*n* = 8)	SG (*n* = 10)	*p*	DG (*n* = 6)	SG (*n* = 9)	*p*
Mean glucose [mmol/L]	7.9 ± 1.6	7.5 ± 1.5	0.5459	7.4 ± 1.5	8.4 ± 1.9	0.3230
SD [mmol/L]	2.5 ± 0.7	1.9 ± 0.5	0.0603	2.9 ± 1.4	2.34 ± 0.4	0.3218
CV [%]	31.3 ± 6.6	26.1 ± 5.0	0.0746	37.5 ± 15.6	29.1 ± 6.8	0.1761
TAR > 13.87 mmol/L [%]	4.2 ± 6.2	0.9 ± 1.6	0.1862	8.9 ± 8.7	3.8 ± 5.6	0.1864
TAR > 10.0 mmol/L [%]	23.5 ± 17.3	15.4 ± 21.8	0.4055	17.3 ± 15.9	25.1 ± 19.6	0.4351
TIR 3.88–10.0 mmol/L [%]	72.3 ± 15.0	81.3 ± 20.9	0.3203	74.1 ± 16.2	70.8 ± 16.7	0.7095
TBR < 3.88 mmol/L [%]	3.9 ± 5.5	2.9 ± 4.0	0.6844	8.2 ± 7.6	3.8 ± 6.3	0.2414

Effects shown as mean ± standard deviations, SD—standard deviation of glycemia, CV—coefficient of variation, TAR—Time above Range, TIR—Time in Range, TBR—Time Below Range.

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
