# Peer review of "Acute Responses to Low and High Intensity Exercise in Type 1 Diabetic Adolescents in Relation to Their Level of Serum 25(OH)D"

_nutrients, 2020, doi:10.3390/nu12020454_

Round 1

Reviewer 1 Report

The article by Mysliwec et al. is a nice paper aiming to establish whether vitamin D status in T1D adolescents affects glycemic control and oxygen consumption during aerobic and mixed (aerobic-anaerobic) physical activity. 

Despite the novelty of the research and the good study design, some major points should be addressed.

1) The majority of the study results are not significant in terms of differences between DG and SG groups. However, it seems there is a trend towards less glycemic variability in the SG compared to DG (assessed by SD and CV) during mixed physical acitivty. This is the finding that should be more emphasized in the discussion, rather than glycemic control assessed by HbA1c.

2) Lines 133-134: Here, and in the Results section, Authors should clarify how much time before: a) each meal and b) the beginning of physical activity was rapid-acting insulin bolus administered. Moreover, Authors should relate the exact amount of rapid-acting insulin administered (bolus, total IU) to the ingested CHO (grams). This is crucial to calculate I:CHO ratio, that would provide, if any, information about possible differences in I:CHO ratio and peripheral insulin sensitivity in-between the 2 study groups.

3) Lines 101-102: Did these patients use sensor-augmented pump (SAP) or closed-loop systems? I assume it was SAP therapy, but it should be specified in the text. Also, was CGM blinded or real-time? Please, specify the device that you used in the study.

4) Lines 106-108: Please, specify if participants on vitamin D supplementatio n were also excluded.

5) Lines 208-210: please, specify here and in the abstract that these differences (HbA1c and DDI) were not significant.

6) Lines 230-231: please, indicate p value.

7) Lines 234-236: please, indicate p-value.

Minor points:

1) Line 48: 0-14 ys range include 5-9 ys range, please rephrase.

2) Lines 49-50: currente evidence suggests that vitamin D deficiency may play a role also in development of islet autoimmunity. Please, rephrase and cite this paper: PMID: 31514368.

3) Line 51: please, update the sentence adding this reference (PMID: 31548248), specifying also that higher serum 25(OH)D levels have been associated with reduced risk of T2D, thus implying a role of vitamin D in pheripheral insulin sensitivity/inflammation

4) Lines 56-58: I would put this sentence out. According to the current clinical and pre-clinical evidence, vitamin D may have a role in T1D prevention mainly through its immunomodulatory and anti-inflammatory effects rather than its effects on pancreatic beta-cell secretory capacity

5) Line 97: Please, put out "normal" and define body weight values as mean +- SD

6) Line 112: Please, define vitamin D deficient group as that including patients with 25(OH)D levels <50 nmol/L, as per guidelines (PMID 21646368)

7) Lines 222-229: Please, cite Table 3 in the text.

Author Response

The article by Mysliwec et al. is a nice paper aiming to establish whether vitamin D status in T1D adolescents affects glycemic control and oxygen consumption during aerobic and mixed (aerobic-anaerobic) physical activity. 

Despite the novelty of the research and the good study design, some major points should be addressed.

Answer: We would like to express our sincere thanks to the Reviewer for the very substantive assessment of our work. All of the remarks are very valuable and will add value to the article.

1) The majority of the study results are not significant in terms of differences between DG and SG groups. However, it seems there is a trend towards less glycemic variability in the SG compared to DG (assessed by SD and CV) during mixed physical acitivty. This is the finding that should be more emphasized in the discussion, rather than glycemic control assessed by HbA1c.

Answer: In the presented study, the researchers have proven that T1DM patients with higher vitamin D levels (SG group) had higher glycemic variability (SD, CV, table 3) during the mixed training than the participants from the DG group. Thus, it may be concluded that glycemic variability is a better parameter characterizing the metabolic control than HbA1c level.

(added; line 323-327).

2) Lines 133-134: Here, and in the Results section, Authors should clarify how much time before: a) each meal and b) the beginning of physical activity was rapid-acting insulin bolus administered. Moreover, Authors should relate the exact amount of rapid-acting insulin administered (bolus, total IU) to the ingested CHO (grams). This is crucial to calculate I:CHO ratio, that would provide, if any, information about possible differences in I:CHO ratio and peripheral insulin sensitivity in-between the 2 study groups.

Answer: All patients ate the same meal (natural yoghurt, oat flakes, banana, walnuts) two hours before every of three exercise tolerance tests. It was composed of 60% of carbohydrates, 15% of proteins and 25% of fat. 40 grams of natural yoghurt, 15 grams of oat flakes, 40 grams of unripe banana and 4 grams of walnuts were included in two carbohydrates units (CU). The amount of CU depended on the total daily caloric requirement of the patient and participants received 0,7 to 0,8 units of insulin for every CU and every fat/protein unit (FPU). (added lines 129-133).

3) Lines 101-102: Did these patients use sensor-augmented pump (SAP) or closed-loop systems? I assume it was SAP therapy, but it should be specified in the text. Also, was CGM blinded or real-time? Please, specify the device that you used in the study.

Answer: Added; Patients using insulin pump therapy integrated with continuous glucose monitoring (CGM). Medtronic electrodes and insulin pumps, Paradigm Veo and MiniMed 640G were used in the study. CGM measured real-time glycaemia and was not blinded, therefore the researchers and the patients had continuous access to glucose concentrations (we added lines 96-99).

4) Lines 106-108: Please, specify if participants on vitamin D supplementatio n were also excluded.

Answer: The participants did not receive any vitamin D supplementation throughout the year before the study and during the study period (we added line 111-113).

5) Lines 208-210: please, specify here and in the abstract that these differences (HbA1c and DDI) were not significant.

Answer: Added; The differences were not statistically significant.

6) Lines 230-231: please, indicate p value.

Answer: Added; in the text

7) Lines 234-236: please, indicate p-value.

Answer: Added in the text.

Minor points:

 1) Line 48: 0-14 ys range include 5-9 ys range, please rephrase.

Answer: Thank you for the valuable remark, we included it in our work. Within the last years, the increase of type 1 diabetes incidence in the pediatric population was stated mainly at the age of 5-9. (added; line 32).

2) Lines 49-50: currente evidence suggests that vitamin D deficiency may play a role also in development of islet autoimmunity. Please, rephrase and cite this paper: PMID: 31514368.

Answer: In the work scientific contributions were included and the article was cited.

3) Line 51: please, update the sentence adding this reference (PMID: 31548248), specifying also that higher serum 25(OH)D levels have been associated with reduced risk of T2D, thus implying a role of vitamin D in pheripheral insulin sensitivity/inflammation

Answer: The phrase was updated by adding the reference (PMID: 31548248):

In addition, Yuan et al. [4] have proved that higher vitamin D concentrations are related with a lower risk of type 2 diabetes development, suggesting that vitamin D plays an important role in increase of insulin sensitivity in patients with carbohydrate metabolism disorder.        

4) Lines 56-58: I would put this sentence out. According to the current clinical and pre-clinical evidence, vitamin D may have a role in T1D prevention mainly through its immunomodulatory and anti-inflammatory effects rather than its effects on pancreatic beta-cell secretory capacity

Answer: Added in the text regarding suggestion

5) Line 97: Please, put out "normal" and define body weight values as mean +- SD

Answer: Added; The body weight in the whole study group was 59.5±12.8 kg and the mean BMI was 20.2±2.6 kg/m2.

6) Line 112: Please, define vitamin D deficient group as that including patients with 25(OH)D levels <50 nmol/L, as per guidelines (PMID 21646368)

Answer: Added; The group of 20 participants enrolled in the study was divided into two subgroups characterized by members with vitamin D deficiency (deficiency group <50 nmol/l, n=10,) and patients with suboptimal vitamin D concentration (suboptimal group 50-75 nmol/l, n=10) [31].

7) Lines 222-229: Please, cite Table 3 in the text.

Answer: Added; (Table 3).

Reviewer 2 Report

With pleasure I read the interesting manuscript entitled "Acute responses to low and high intensity exercise in 2 type 1 diabetic adolescents in relation to their level of 3 serum 25(OH)D" by Mysliwec et al., which deals with the most common non-infectious chronic disease in the young population in Europe. I found the manuscript written well, in a clear and exhaustive manner. The conclusions of the authors may represent an important hint for the evaluation of biochemical factors implicated in the metabolic balance maintenance in response to physical activity in the Diabetes mellitus type 1 affected patients. I consider the manuscript and the conclusions of it very interesting to the readers and researches in this field. I consider the manuscript ready to be published in the present form.

Author Response

With pleasure I read the interesting manuscript entitled "Acute responses to low and high intensity exercise in 2 type 1 diabetic adolescents in relation to their level of 3 serum 25(OH)D" by Mysliwec et al., which deals with the most common non-infectious chronic disease in the young population in Europe. I found the manuscript written well, in a clear and exhaustive manner. The conclusions of the authors may represent an important hint for the evaluation of biochemical factors implicated in the metabolic balance maintenance in response to physical activity in the Diabetes mellitus type 1 affected patients. I consider the manuscript and the conclusions of it very interesting to the readers and researches in this field. I consider the manuscript ready to be published in the present form.

Answer: We would like to express our sincere thanks to the Reviewer for the excellent assessment of our article, in which we presented the problems of patients with type 1 diabetes. As far as to our knowledge, this is one of the very few studies described in the literature in this regard. Certainly, we will continue our work on a larger population of adolescents and adults.

Reviewer 3 Report

Few studies examined whether vitamin D deficiency is associated with glycemic variability during exercise in young type I diabetic (T1D) patients.  This study aimed to fill the gap and thus is significant. The authors measured oxygen consumption and glucose levels during aerobic and mixed aerobic and anaerobic exercise in T1D patients with vitamin D deficient or suboptimal levels of vitamin D. The authors reported that vitamin D deficiency is associated with higher HbA1C levels and greater variability of blood glucose concentrations, but not oxygen consumption. The presentation is clear. However, the conclusions were based upon insignificant results, likely due to small patient numbers.

Author Response

Few studies examined whether vitamin D deficiency is associated with glycemic variability during exercise in young type I diabetic (T1D) patients.  This study aimed to fill the gap and thus is significant. The authors measured oxygen consumption and glucose levels during aerobic and mixed aerobic and anaerobic exercise in T1D patients with vitamin D deficient or suboptimal levels of vitamin D. The authors reported that vitamin D deficiency is associated with higher HbA1C levels and greater variability of blood glucose concentrations, but not oxygen consumption. The presentation is clear. However, the conclusions were based upon insignificant results, likely due to small patient numbers.

Answer: We would like to express our sincere thanks to the Reviewer for the remarks regarding our work. There is indeed a small number of researches concerning the research issues, that we studied in our article. Particularly, it regards the reaction of patients with type 1 diabetes to the physical effort of low, high or mixed intensity. Moreover, there is no evidence concerning the simultaneous analysis of vitamin D concentrations in patients. The issue regarding vitamin D, that we raised in our work, is especially essential in the countries where vitamin D deficiency is registered. The small number of participants may have been one of the reasons for the lack of statistical significance between the studied groups. Therefore, we plan to continue our work soon. The studies are very difficult from the logistic point of view and it is not easy to find a suitable group of participants to our study. The final version of our work was edited regarding the description of the project, research methods and the conclusions, in order to improve its transparency.

Round 2

Reviewer 1 Report

Overall, authors addressed the majority of comments.

However, authors state that individuals with higher vitamin D levels had higher glycemic variability during the mixed training compared to DG group (lines 330-334). By looking at Table 3, mean SD and mean CV (markers of glycemic variability) are both higher in the DG group compared to SG group during both mixed and aerobic training. Therefore, this sentence has to be rephrased. Also, the statement that glycemic variability is a a better parameter to assess metabolic control compared to A1c should be put out, because it is a strong statement based on such small sample size.

Line 109: please, use the term "sensor" in place of "electrodes".

Lines 50-53: please put out the two sentences from "What is more [..] until "increase of insulin sensitivity". They are redundant and do not refer to the previously cited paper.

Author Response

 However, authors state that individuals with higher vitamin D levels had higher glycemic variability during the mixed training compared to DG group (lines 330-334). By looking at Table 3, mean SD and mean CV (markers of glycemic variability) are both higher in the DG group compared to SG group during both mixed and aerobic training. Therefore, this sentence has to be rephrased. Also, the statement that glycemic variability is a a better parameter to assess metabolic control compared to A1c should be put out, because it is a strong statement based on such small sample size.

 Answer; Corrected; In the presented study, the researchers have proven that T1DM patients with higher vitamin D levels (SG group) had higher lower glycemic variability (SD, CV, table 3) during the mixed and aerobic training than the participants from the DG group. Thus, it may be concluded that glycaemic variability is a better parameter characterizing the metabolic control than HbA1c level.

Line 109: please, use the term "sensor" in place of "electrodes".

 Answer; Corrected; Patients using insulin pump therapy integrated with continuous glucose monitoring (CGM). Medtronic electrodes sensor and insulin pumps, Paradigm Veo and MiniMed 640G were used in the study. CGM measured real-time glycaemia and was not blinded, therefore the researchers and the patients had continuous access to glucose concentrations.

Lines 50-53: please put out the two sentences from "What is more [..] until "increase of insulin sensitivity". They are redundant and do not refer to the previously cited paper.

Answer; Corrected; …diabetes and there is evidence suggesting that vitamin D deficiency may play a role in the development of pancreatic beta cells autoimmunity [3]. What is more, in the same study, researchers have shown that higher 25(OH) D level is associated with a lower risk of type 2 diabetes development. It has been suggested that vitamin D has a pivotal role in increase of insulin sensitivity. In addition, Yuan et al. [4] have proved that higher vitamin D concentrations are related with a lower risk of type 2 diabetes development, suggesting that vitamin D plays an important role in increase of insulin sensitivity in patients with carbohydrate metabolism disorder.